# Immunogenicity and protective efficacy of SARS-CoV-2 mRNA vaccine encoding secreted non-stabilized spike in female mice

Establishment of an mRNA vaccine platform in low- and middle-income countries (LMICs) is important to enhance vaccine accessibility and ensure future pandemic preparedness. Here, we describe the preclinical studies of "ChulaCov19", a SARS-CoV-2 mRNA encoding prefusion-unstabilized ectodomain spike protein encapsulated in lipid nanoparticles (LNP). In female BALB/c mice, ChulaCov19 at 0.2, 1, 10, and 30 μg elicits robust neutralizing antibody (NAb) and T cell responses in a dose-dependent relationship. The geometric mean titers (GMTs) of NAb against wild-type (WT, Wuhan-Hu1) virus are 1,280, 11,762, 54,047, and 62,084, respectively. Higher doses induce better cross-NAb against Delta (B.1.617.2) and Omicron (BA.1 and BA.4/5) variants. This elicited immunogenicity is significantly higher than those induced by homologous CoronaVac or AZD1222 vaccination. In a heterologous prime-boost study, ChulaCov19 booster dose generates a 7-fold increase of NAb against Wuhan-Hu1 WT virus and also significantly increases NAb response against Omicron (BA.1 and BA.4/5) when compared to homologous CoronaVac or AZD1222 vaccination. Challenge studies show that ChulaCov19 protects human-ACE-2-expressing female mice from COVID-19 symptoms, prevents viremia and significantly reduces tissue viral load. Moreover, anamnestic NAb response is undetectable in challenge animals. ChulaCov19 is therefore a promising mRNA vaccine candidate either as a primary or boost vaccination and has entered clinical development.

Since COVID-19, the disease caused by severe acute respiratory virus 2 (SARS-CoV-2), began to spread in late December 2019, it has since become a global pandemic[1]. Even though most COVID-19 patients are asymptomatic or only mildly symptomatic[2–4], the virus is still eminently transmissible even during the early phases of the illness. This contrasts with SARS CoV-1 where peak viral shedding occurs after patients were already quite ill[5,6]. Such unusual characteristics, in conjunction with a highly contagious profile, resulted in the rapid spreading of the virus worldwide. In just over 2 years into the pandemic, more than 10 variants of the virus have been reported, of which 5 variants, including Alpha (B.1.1.7), Beta (B.1.351), Gamma (P.1), Delta (B.1.617.2), and Omicron (B1.1.529) have been categorized by WHO as

variants of concern (VOCs)[7]. These viruses adapted to increase the transmissibility, severity and/or immune evasion[8]. By 18th August 2022, almost 600 million confirmed cases were caused by multiple VOCs and almost 6.5 million deaths were reported[9]. Presently, the pandemic is still surging in many countries.

Currently there are at least 11 approved vaccines using various technology platforms, including mRNA, inactivated virus, viral-vector and recombinant protein[10]. The vaccine effectiveness is varied due to several factors such as the emergence of new variants, study population, and prevalence of the outbreak during the period the studies were conducted[11–13]. Although the currently available vaccines do not completely prevent infection, they are efficacious in reducing severe

✉e-mail: chutitorn.k@chula.ac.th

symptoms of infected individuals[11]. Unfortunately, it has also been proven that vaccine efficacy decreases over time[14]. Together with the emergence of new VOCs, a booster dose (either homologous or heterologous vaccine modality) is required to enhance the vaccine effectiveness[15].

Among the recently approved vaccines, mRNA modality seems to be the most efficacious as it induces high levels of desired immune responses and protects from severe symptoms[16,17]. Moreover, the feasibility of large-scale production as well as rapid adaptability to new variants are major advantages of the mRNA production platform. The spike (S) protein of the virus, which contains the major neutralizing epitopes in the receptor binding domain (RBD) and N-terminal domain (NTD), has proven to be the most promising immunogen[18]. Thus, most recently approved vaccines employ full-length S (with or without modification) or whole virus (inactivated) as a target antigen[19].

Vaccine inequity issue remains a major global challenge. Broad and timely access to effective vaccines in LMICs, particularly the most under-served settings, has always been limited during past pandemics and this has extended to COVID-19[20]. Developing highly effective vaccine platforms like mRNA technology in low- and middle-income countries (LMICs) is therefore an important goal[21]. In response to the COVID19 pandemic and in preparation for future pandemics, Thailand has funded this mRNA vaccine development program from preclinical to manufacturing and clinical development. Here, we describe the construction and preclinical evaluation of mRNA expressing the ectodomain of native, prefusion-non-stabilized S protein of wild-type (WT) Wuhan-Hu1 strain encapsulated within lipid nanoparticles, henceforth referred to as ChulaCov19. The vaccine was measured for its immunogenicity in BALB/c mice both using ChulaCov19 alone or as heterologous prime/boost regimens alongside the approved vaccines (Fig. 1a). It was also evaluated for the protective efficacy in transgenic mice expressing human angiotensin-converting enzyme-2 (ACE2), Fig. 1b.

## Results

### In vitro protein expression and particles characterization

The purified mRNA-S (ChulaCov19) with undetectable endotoxin was tested for protein expression in VERO E6 cells. By using immunofluorescent assay, employing RBD-, S1-, S2-specific antibodies or PCS, the S proteins were observed within the cytoplasm of transfected cells while untransfected cells were negative for fluorescent signal (Fig. 2a). Using western blot, the S protein could be detected in cell culture supernatant when using anti-RBD, -S1, -S2 and PCS as primary antibodies. The bands corresponding to S1, S2 and intact S (S0) were detected. The comparable molecular weight of S0 expressed by ChulaCov19 was also observed when using commercial recombinant S with S1/S2 cleavage site abolished as control (Fig. 2c). The function of secreted S protein also determined whether it could bind to hACE-2. Signals of S protein stained by RBD-, S1-, S2-specific antibodies or PCS were detected on unpermeabilized HEK293T-hACE-2 cell after incubation with transfected supernatant. The results resembled those observed in the panel that used a commercial recombinant S-trimer instead of transfected supernatant. In contrast, undetectable fluorescent signals for S proteins were observed when HEK293T-hACE-2 were incubated with supernatant from untransfected cells (Fig. 2b).

The encapsulated mRNA-LNP was characterized by various parameters including size, polydispersity (PDI) and mRNA encapsulation efficiency at 1, 6, and 12 months after manufacture. The results demonstrated that, at least up to 12 months, only minor changes were observed when the particles were stored in −75 °C (Supplementary Table 1) and were still within the acceptable criteria.

### Spike-specific total IgG antibody analysis of one versus two doses of immunization

The S-specific total IgG after 1 or 2 doses of ChulaCov19 was analyzed in mice sera from experiment 1. S-specific total IgG analyzed at week 2

revealed that all ChulaCov19-immunized mice, either with 1 or 2 doses, elicited anti-S-specific IgG response from the lowest dose of 0.2 μg with a dose-dependent response pattern. The second dose of ChulaCov19 strongly augmented the IgG antibody levels with an increase of 10-19 folds, $p < 0.01$ for all dose ranges (Fig. 3a). IgG2a and IgG1 subclasses were also assessed to determine Th1 and Th2 responses, respectively. The results demonstrated that IgG2a/IgG1 (or Th1/Th2) ratios were greater than 1 in all vaccinated mice (Fig. 3b). These results reflect that ChulaCov19 was highly immunogenic and induced a Th1-skewed response in mice.

### Neutralizing antibody results

NAb measurements in mice sera from Experiment 1 against WT (Wuhan-Hu1) live-virus (micro-VNT50) at 2-week after each dose showed NAb response in a dose-dependent manner. After the first dose, NAb were detected in mice that received 1, 10, and 30 μg ChulaCov19 with corresponding GMTs of micro-VNT50 titer of 80, 368, and 735, respectively. The NAb titers were drastically enhanced after the second dose was given, $p < 0.01$ for all dose ranges. After 2-dose, the GMTs of micro-VNT50 titer for 0.2, 1, 10, and 30 μg were 1280, 11,763, 54,047, and 62,084, respectively (Fig. 4a). This finding implied that ChulaCov19 is highly immunogenic against WT (Wuhan-Hu1) strain. Mice sera were further analyzed for NAb by psVNT50 test against the important recent VOCs, including Delta (B.1.617.2) variant and Omicron (BA.1 and BA.4/5) variants, and titers significantly decreased for all VOCs. For example, for 10 μg dose group, the GMTs of psVNT50 for Delta (B.1.617.2) and Omicron (BA.1) variants decreased 5.9 and 14.3 folds when compared against WT (Wuhan-Hu1) strain, respectively (Fig. 4b). Meanwhile, psVNT50 against BA.4/5 subvariant showed the lowest GMT in 1, 10, and 30 μg dosed groups. When compared with psVNT50 titers against BA.1, the GMT reduction against BA.4/5 in 10 and 30 μg dosed groups were 48 and 2.3 folds, respectively. This result implied that the decrease in Nab titers against BA.4/5 may be improved with higher mRNA vaccine doses.

ChulaCov19 was further compared to two approved vaccines (CoronaVac and AZD1222), either in a homologous prime/boost setting or heterologous one (i.e. as a booster dose in mice that had been primed with CoronaVac or AZD1222 (Experiment 2). The same dosage of approved vaccines were used with a dose of 5 μg ChulaCov19 (1/10 of the human dose used in Phase 2 Trial). For the homologous prime/boost, ChulaCov19 showed 3- to 10.6-fold higher NAb levels compared to 2-dose immunization of CoronaVac or AZD1222 across all variants WT (Wuhan-Hu1), Alpha (B.1.1.7), Beta B.1.351), and Delta (B.1.617.2), as measured by micro-VNT50 (Fig. 4c). For the heterologous prime/boost, mice primed with CoronaVac or AZD1222 and then boosted with ChulaCov19 generated significantly higher GMT against WT (Wuhan-Hu1), Alpha (B.1.1.7), Beta (B.1.351), Delta (B.1.617.2), and Omicron (B.1.529) when compared to the respective homologous prime/boost groups. For example, the micro-VNT50 GMT against WT (Wuhan-Hu1) in the AZD1222-prime/ChulaCov19-boost group was 7-fold higher than 2-dose AZD1222 immunization (GMT of micro-VNT50 were 31,042 vs 4457, $p = 0.0079$). And the GMT NAb titer against WT (Wuhan-Hu1) in the CoronaVac-prime/ChulaCov19-boost group was also 7-fold higher than 2-dose of the CoronaVac group (GMT of micro-VNT50 were 23,525 vs 3378, $p = 0.0317$), Fig. 4c. In the case of Omicron variants, psVNT50 NAb GMT results against Omicron BA.1 and BA.4/5 subvariants showed that the heterologous prime/boost regimen was more efficient (84-172 folds increase) in inducing NAb against BA.1 and BA.4/5 subvariants compared to homologous CoronaVac or AZD1222 immunization. For example, the psVNT-50 against BA.1 in the CoronaVac-prime/ChulaCov19-boost group (psVNT-50 GMT = 875) was significantly higher ($p < 0.01$) than homologous CoronaVac (psVNT-50 GMT = 5.1) and homologous AZD1222 (psVNT-50 GMT = 2.7) groups. Similar findings were also observed in BA.4/5 subvariant (Fig. 4d). These results confirmed that ChulaCov19 is highly immunogenic either

**a** **Experiment 1: Dose-response study of homologous ChulaCov19 prime/boost immunization**

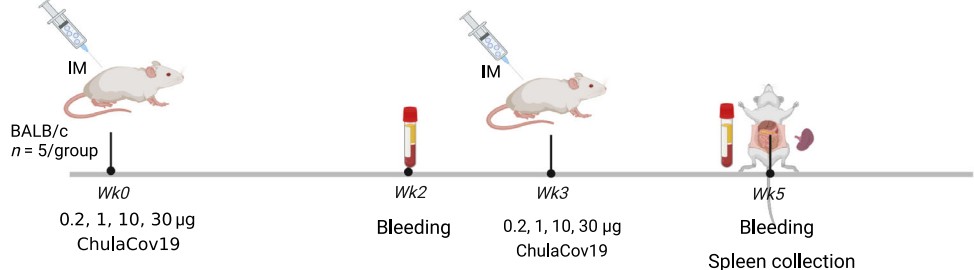

**Experiment 2: Prime/boost regimen of ChulaCov19 and approved vaccines**

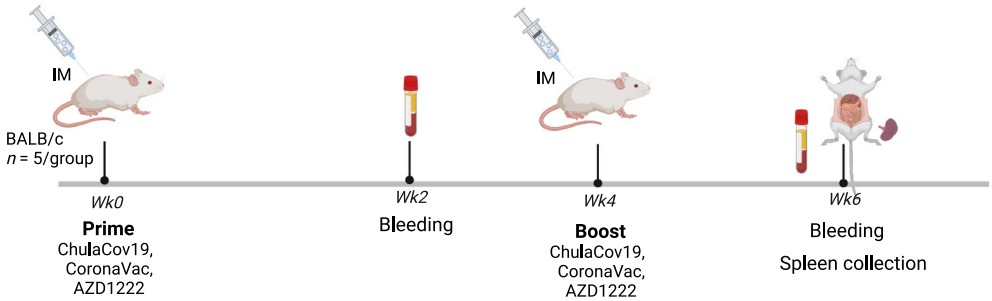

**Experiment 3: Antibody durability and effect of 3ᵈ dose of ChulaCov19**

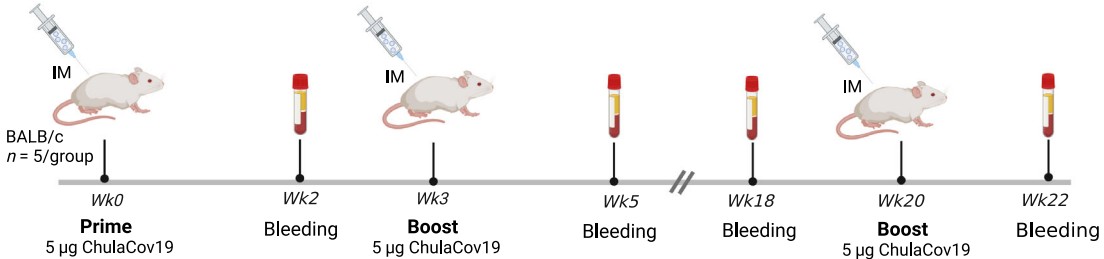

**b** **Challenge study**

as a primary vaccination in a vaccine-naïve setting, or as a booster vaccine in animals previously vaccinated with other vaccines.

In Experiment 3, the durability of NAb induced by ChulaCov19 was monitored until week 18 (15 weeks after the 2nd dose). The results revealed that the NAb against WT (Wuhan-Hu1) and Delta (B.1.617.2) variants were still detectable in all mice (5/5) but 4/5 mice for Omicron BA.1 and BA.4/5. At week 18, the NAb against WT (Wuhan-Hu1) and

Delta (B.1.617.2) decreased approximately 2-fold but not statistically significant when compare with week 5 titers. At this time-point, the NAb titers against both Omicron subvariants were still in the same level with week 5 titers (Fig. 4e). This implies that ChulaCov19 could induce a long-lasting NAb, at least until 15 weeks postimmunization especially against WT (Wuhan-Hu1) and Delta (B.1.617.2) variants. Interestingly, the 3rd dose of ChulaCov19 administered at 17-week apart significantly

**Fig. 1 | Immunization schedule of ChulaCov19 in BALB/c mice. a** Experiment 1: mice were immunized twice intramuscularly (IM) with a 3-week interval with various dosages of ChulaCov19 at 0.2, 1, 10 and 30 μg. Experiment 2: heterologous prime-boost study, mice were primed with 1/10 of the approved human dosage of CoronaVac or AZD1222 and boosted 4 weeks later with 5 μg of ChulaCov19. Homologous prime/boost of each vaccine (CoronaVac, AZD1222, or ChulaCov19) were included as control groups. Experiment 3: antibody durability and effect of 3rd dose of ChulaCov19 study, mice were immunized twice with 3 weeks interval with 5 μg of ChulaCov19 (1/10 of human dose used in clinical trial) then boosted again at week 20. Bleeding was performed at 2 weeks following each dose (and at week 18 for Experiment 3). Splenocytes were collected at 2 weeks after the second dose (Experiment 1 & 2). *n* = 5 per group for Experiment 1, 2 and 3. **b** Challenge study in K18-hACE2 transgenic mice, *n* = 6 in vaccinated groups and n = 5 in control (PBS-receiving) group. Animals were immunized IM with 1 μg or 10 μg of ChulaCov19 at weeks 0 and 3. Sera were collected at weeks 0, 2, 3, 4 + 6 days, and 5 + 6 days for NAb measurements. At week 5, mice were challenged intranasally with 2×10⁴ pfu of WT SARS-CoV-2. Tissues were collected at week 5 + 6 days for assessment of viral RNA. Figures were created with BioRender.com.

boosted the NAb against all variants analyzed. At week 22, the psVNT-50 GMT for WT (Wuhan-Hu1), Delta (B.1.617.2), BA.1 and BA.4/5 were 25,539, 10,722, 2133, and 1707, respectively; 13-57 folds increase from the pre-boost baseline (Week18).

## SARS-Cov-2-spike specific T cell responses

Splenocytes from mice immunized with various dosages of Chula-Cov19 (Experiment 1) were analyzed as summed frequency of S-specific IFN-γ positive T cells (Fig. 5a). Similar to the antibody results, the magnitude of T cell response was found to be dose-dependent but peaking at the 10-μg dosage. However, the slightly higher level compared to the 30-μg group was not statistically significant. Mean spike-specific IFN-γ positive T cells for 0.2, 1, 10 and 30 μg were 166, 429, 1913, and 1378 SFC/10⁶ splenocytes, respectively. T-cell responded to S1-pooled peptides much more common than to S2-pooled peptides. The analysis of the responses to different parts of S-specific pool peptides in all vaccinated groups showed that peptide pool #3-5 (which include receptor-binding domain or RBD) and pool #9 (which includes Heptad Repeat 2 or HR2) in S1 and S2, respectively, were the most common peptides pools recognized by the vaccinated mice T-cells. In all vaccinated groups, the number of spots that were detected after peptide pool #3-5 and pool #9 stimulation were 74–84% and 8–10%, respectively (Fig. 5a).

In the heterologous *vs* homologous prime/boost experiment (Experiment 2), homologous ChulaCov19 and homologous AZD1222 immunizations elicited comparable levels of S-specific IFN-γ positive T cells responses which was 2482 and 2210 SFC/10⁶ splenocytes, respectively. Of interest, the heterologous AZD1222-prime/Chula-Cov19-boost induced the best specific T cells responses with mean spike-specific IFN-γ positive T cells of 3725 SFC/10⁶ splenocytes, which approximately 1.7-fold higher than homologous ChulaCov19 (*p* = 0.1934) and also significantly higher than other groups (*p* < 0.05). In contrast, CoronaVac immunization showed the lowest T cells responses (42 SFC/10⁶ splenocytes). Boosting with ChulaCov19, although not statistically significant, it could enhance the IFN-γ positive T cells by approximately 6.5 folds (*p* = 0.1523) of the magnitude of T cells response in CoronaVac-primed mice (273 SFC/10⁶ splenocytes). However, this was still far lower than using homologous ChulaCov19 or AZD1222-prime/ChulaCov19-boost immunization regimens (Fig. 5b).

## Immunogenicity and protective efficacy of ChulaCov19 in K18-hACE2 transgenic mice

After 2 doses of ChulaCov19 or phosphate-buffered saline (PBS, control group) with a 3-week interval, K18-hACE2 mice were tested for NAb kinetics against live SARS-CoV-2 strain hCoV-19/Hongkong/VM20001061/2020. Baseline NAb levels at week 0 of all mice were negative. However, at week 2 after the first dose, 6/6 and 4/6 animals from the 10 μg and 1 μg groups, respectively, showed a dose-dependent manner of NAb response to vaccine administration. At this time-point, 10 μg dosed mice induced significantly higher in GMTs of micro-VNT50 titers than 1 μg dosed mice (*p* = 0.0065). At week 3 after dose 1, NAb were still detected in all animals in the 10 μg group, and 5/6 animals in the 1 μg group. At week 5 (2 weeks after the second dose), all mice in both vaccinated groups showed increased NAb levels.

The GMT of micro-VNT50 titers at week 5 were 15,343 and 4424 in the 10 μg and 1 μg groups, respectively, *p* = 0.0325. Day 6 after the viral challenge (week 5 + 6 days), there was a slight decline of NAb titers in both groups but not statistically significant when compared to week 5, *p* = 0.1126 and *p* = 0.4437 for 10 μg and 1 μg groups, respectively. In contrast, sham-treated animals failed to show any NAb response except for one animal on Wk5 + 6d (Fig. 6a). Of note, at week 5, all vaccinated mice at the 10 μg dose, and 5 of 6 mice at 1 μg dose elicited SARS-CoV-2 specific serum IgA (supplementary Figure S1a and S1b). There were no anamnestic responses (four-fold increase on micro-VNT50 titers) in all vaccinated groups 6 days after the challenge, whereas one mouse in the control group developed a low micro-VNT50 titer at 40.

After SARS-CoV-2 challenge, there was no measurable decline in body weight among vaccinated groups. The average decline from peak to euthanasia among PBS-receiving mice was 17%. The average body weight by group from week 5 to week 5 + 6 days was demonstrated in Fig. 6b. By Day 4 after challenge, two mice in PBS-receiving group (control) began to show clinical signs of anorexia, lethargy, and rough hair coat. On Day 5, significant weight reduction (*p* < 0.05) was observed in control group when compared with the vaccinated groups. Moreover, all five mice in control group exhibited varying symptoms of increased anorexia, lethargy, immobility, rough hair coat and increased respiration rate and effort. Three out of five mice reached euthanasia criteria on Day 5, and symptoms progressed for the remaining two mice which met the criteria on Day 6. In contrast, mice that received 2 doses of either 1 or 10 μg of ChulaCov19 were normal with no symptoms throughout postchallenge period of 6 days.

The RT-qPCR data showed that both doses of vaccine prevented the expression of SARS-CoV-2 viremia at 5 or 6 days after viral inoculation. There was no detectable viremia in mice in both high or low-dose vaccine-treated groups while an average of 7.71×10⁴ GE/mL (ranged from 1.03×10³ – 3.75×10⁵ GE/mL) of viral RNA was detected in PBS-received mice, Fig. 6c. These results suggest that both dosing regimens effectively protected the mice from detectable levels of circulating virus. Moreover, the low dose regimen was also shown to induce a marked reduction in viral load in nasal turbinates, brain, and lung tissues compared to sham-treated controls. The average reduction of viral load in tissues of both vaccine-treated groups relative to the control was 99.9-100%.

In the lung, inflammation was limited to predominantly peribronchiolar proliferation of mononuclear cells, akin to an expansion of cellularity among bronchiolar lymphoid tissue but without notable follicle formation. In the nasal turbinate, vaccinated mice exhibited luminal accumulation of mucus and/or fibrin, albeit only minimal to mild amounts. Immunofluorescent results mostly correlate with PCR data. No positive detection of viral RNA was present in the 10 μg group of animals analyzed by ISH. Among the 1 μg group, only one tissue had very few positive cells, the nasal epithelium. It is notable that while all mice, except for one, dosed with 10-μg and 1-μg ChulaCov19 showed no detectable SARS-CoV-2 viral RNA in tested tissues. In the control group, positive viral RNA staining was present in individual neurons of the olfactory bulb (4/4), epithelial cells of the nasal sinus (4/5), alveolar epithelial cells and macrophages in the lung (5/5), see Table 1.

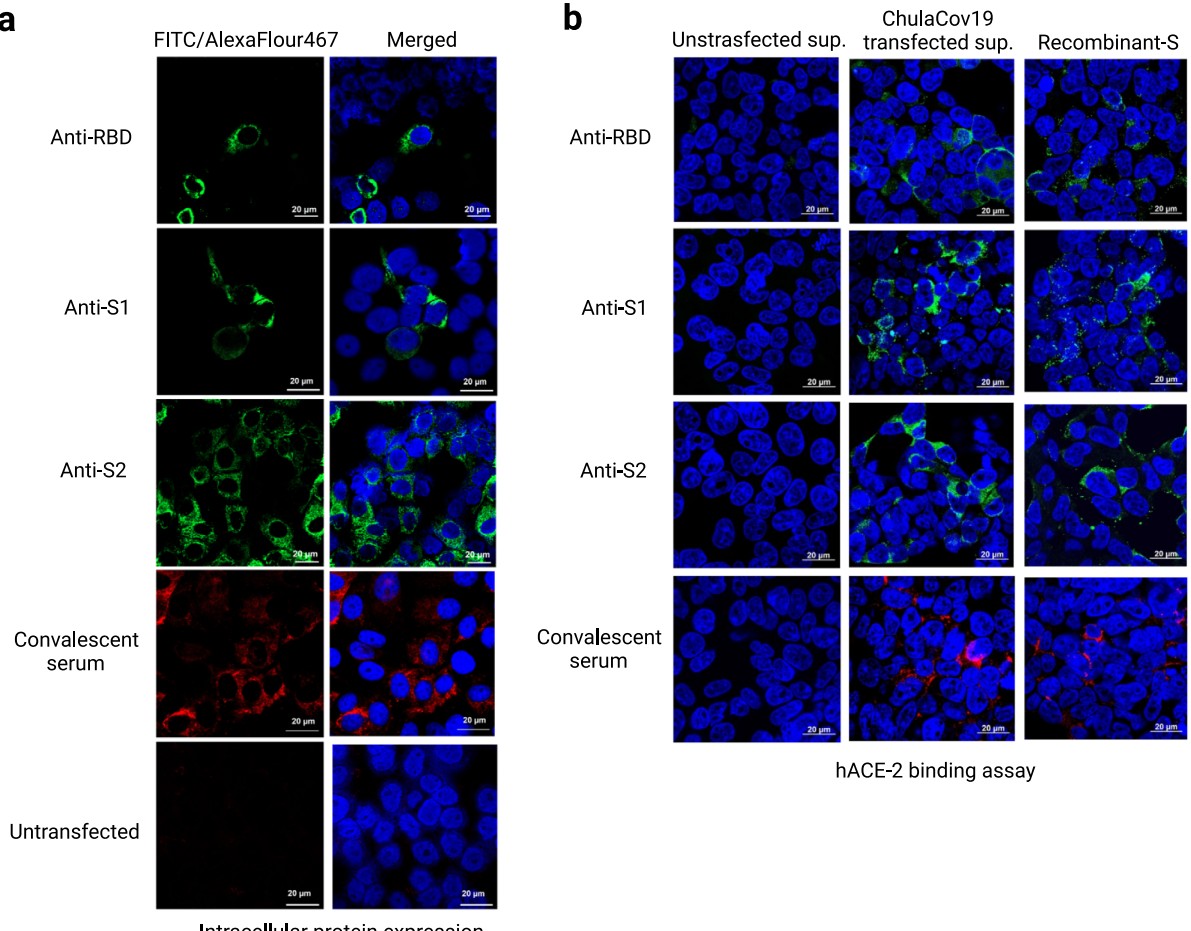

Intracellular protein expression

hACE-2 binding assay

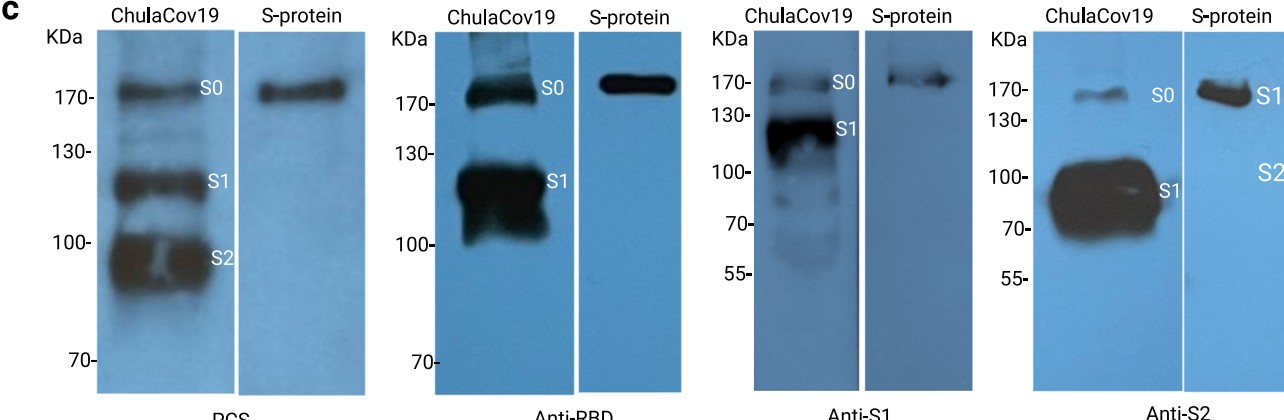

**Fig. 2 | Protein expression analysis at 24 h after mRNA transfection in VERO E6 cells. a** Intracellular S protein expression examined by immunofluorescent assay employing anti-RBD, -S1, -S2 or PCS as primary antibody, the nuclei were counter stained with DAPI (blue). FITC-tagged 2$^{nd}$ Abs (green) were used for detection of RBD, S1, and S2 while AlexaFluor647-tagged 2$^{nd}$ Ab (red) was used following PCS staining. **b** hACE-2 binding assay (merged): culture supernatant collected from ChulaCov19 transfected cells incubated with HEK293T- hACE-2 cells. S protein on HEK293T-hACE-2 cell surface was stained with the same antibodies used in 2a. **c** S protein expression in cell culture supernatant analyzed by western blot using anti-RBD, -S1, -S2 or PCS as primary antibody. Recombinant S protein with abolished S1/S2 cleavage site was used as positive control in HEK293T-hACE-2 binding assay (right panel of 2**b**) and western blot (right lane of each panel in 2**c**). S0 was used to depict unprocessed S protein. Experiments were repeated two times independently with similar results.

## Discussion

In this study, ChulaCov19 was shown to be highly immunogenic, in a dose-responsive relationship, even when immunized with very low amount of 0.2 μg as measured by both live- and pseudovirus-neutralization assays. Protection against WT (Wuhan-Hu1) viral challenge in K18-hACE2 transgenic mice mediated by ChulaCov19 was successfully demonstrated. At 2×10$^4$ PFU of SARS-CoV-2 inoculum, PBS-vaccinated mice displayed clinical symptoms or weight loss within 1 day and all mice succumbed by day 6. This was consistent with the prior study in K18-hACE2 that intranasal inoculation with the similar range of virus caused death within 1 week[22]. In contrast, ChulaCov19 immunized mice, both 1 μg and 10 μg doses enabled 100% survival

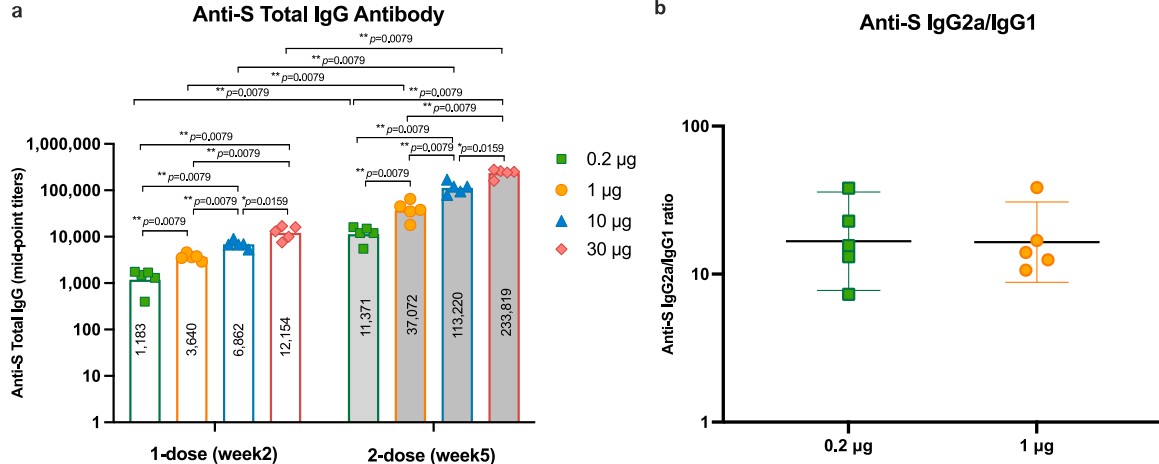

**Fig. 3 | S-specific IgG responses in ChulaCov19 immunized mice. a** Kinetics of total IgG at 2 weeks after receiving 1 or 2 doses of 0.2, 1, 10, and 30 μg of Chula-Cov19. **b** S-specific IgG2a/IgG1 ratio measured at 2 weeks after the 2nd dose. Note; the IgG2a/IgG1 ratio of 10 μg and 30 μg immunized mice were not analyzed due to limited volume of serum samples. The mid-point titers were determined in duplicate assays from 5 mice in each group. Bars (**a**) or horizontal lines (**b**) represent the geometric mean (GMT) for each group while error bars indicate the 95% confident interval. Each dot represents an individual animal. Statistical analysis significance was determined by two-sided Mann–Whitney test. Differences were considered significant at $p < 0.05$ with exact $p$-values shown. $p < 0.05$ and $p < 0.01$ are indicated by * and **, respectively. Source data are provided as a Source Data file.

compared to full mortality rate in PBS-immunized mice. This demonstrated the significant protective efficacy of ChulaCov19 in the preclinical phase.

K18-hACE2 transgenic mice are highly susceptible and displayed clinical signs following SARS-CoV-2 challenge[22,23]. Body weight of ChulaCov19 vaccinated mice decreased slightly only at days 1 and 2 post-challenge then gradually increased to the same levels as pre-challenge at day 6 (Fig. 6b). This was in accordance with previous studies showing that survived animals, either using HFH4-hACE2 or K18-hACE2 strains, could recovery their body weight to the basal levels at pre-challenge while weight lost continued for unprotected or non-vaccinated animals and reached euthanized criteria within approximately 1 week[24–26]. In addition, there was no anamnestic antibody response detected in the ChulaCov19 vaccinated mice after viral challenge (Fig. 6a). This is a surrogate marker indicative of vaccine effectiveness, or the sterilizing immunity as reported in the previous study[27]. Notably, SARS-CoV-2 RNA measured by ISH was undetected in lung tissues in mice vaccinated with ChulaCov19 at either 1 or 10 μg dose. When RT-qPCR was used, although viral RNA was still detected in some tissues, both dosages demonstrated a 99-100% reduction of viral RNA in tested tissues when compared to the control group. Similar with the previous study, low level of viral RNA occasionally detected in survived mice was also reported by studies that used K18-hACE2 as a model[28]. The possible explanation of the higher detectable viral RNA found in 10 μg compared to 1 μg immunized mice (Fig. 6c) may be due to RT-qPCR, a highly sensitive method detecting the free viral RNA from disintegrated virus. Further investigation using different techniques, such as viral isolation and titration from the collected tissues is required to draw a definite conclusion. Beyond the techniques used for the viral detection, the inverse correlation between vaccine dosage and tissue viremia might be the results of the quality of T cell response induced by the high vaccine dosage. Previous studies reported that low-dose vaccination induced only high avidity T cells. In contrast, a higher dose vaccination not only induced the mixture of low and high avidity T cells responses, but also induced the clonal deletion of high avidity CD8 T cells[29–31]. More importantly, according to the mechanism demonstrated by Derby M, et al., high avidity T cells could recognize and clear virus-infected cells more rapidly than low avidity T cells as it requires a small amount of viral antigen. On the contrary, low avidity T cells which require a higher amount of viral antigen were able to lyse

the viral infection after the new virion were produced[31]. Hence, in this study, although the NAb was displayed in a dose-dependent fashion, in-depth analysis of T cell quality induced by different vaccine dosage is also needed to investigate the controversy of viremia after challenge.

When correlating protective efficacy and NAb titers induced by ChulaCov19, a micro-VNT50 titer of 2,560 before challenge in 1 μg immunized mice was found to completely prevent viral burden in the lung as analyzed by ISH and RT-qPCR (Figs. 6b, c, Table 1). This is similar to the previous study of mRNA-1273, which demonstrated that a minimum NAb titer (analyzed by focus reduction neutralization test) of approximately 2,000 was required to completely protect K18-ACE2 mice from ancestral virus with D614G infection[32].

Regarding the vaccine construct characterization, protein expression studies revealed S proteins were expressed both in intracellular and extracellular compartments when detected either by specific antibodies or patient sera (Fig. 2a). In supernatant, we could detect both intact S and cleaved S1 and S2 (Fig. 2c). These results reflect the real S protein dynamic as shedding of S1 could be detected in viral infection[33,34]. Function of the expressed S protein was also confirmed as it could bind to hACE-2 similar to those of stabilized trimeric spike (Fig. 2b). Although several SARS-CoV-2 vaccines used an engineered S protein to abolish S1/S2 cleavage or to stabilize the prefusion stage[35–37], vaccines encoding unmodified S protein are also worth exploring as its structure is the same as native viral protein. Moreover, ChAdOx1: AZD1222 that used unmodified S has been shown to induce high level of NAb and T cells responses even after a single immunization dose in two mouse strains[38]. In the same study, two doses of AZD1222 could protect rhesus macaque form viral challenge. In addition, AZD1222 was also showed to be effective in clinical trials[39,40]. The structural study of S protein expressed by AZ1222 showed a native-like structure mostly found in the prefusion stage[41]. Post-translational modifications were also similar to those observed on SARS-CoV-2[41]. However, further beneficial evaluation on the use of native-like S protein structure requires in-depth analysis in clinical settings especially in immune elicitation characteristics.

A Th2 dominant response following the vaccination remains a major concern of immunopathology that caused lung inflammation as reported in respiratory syncytial virus (RSV), SARS-CoV-1 and MERS-CoV[42–45]. Previous study by Eichinger KM, et al. demonstrated that only Th2-dominant but not Th1/Th2 balanced response enhanced lung

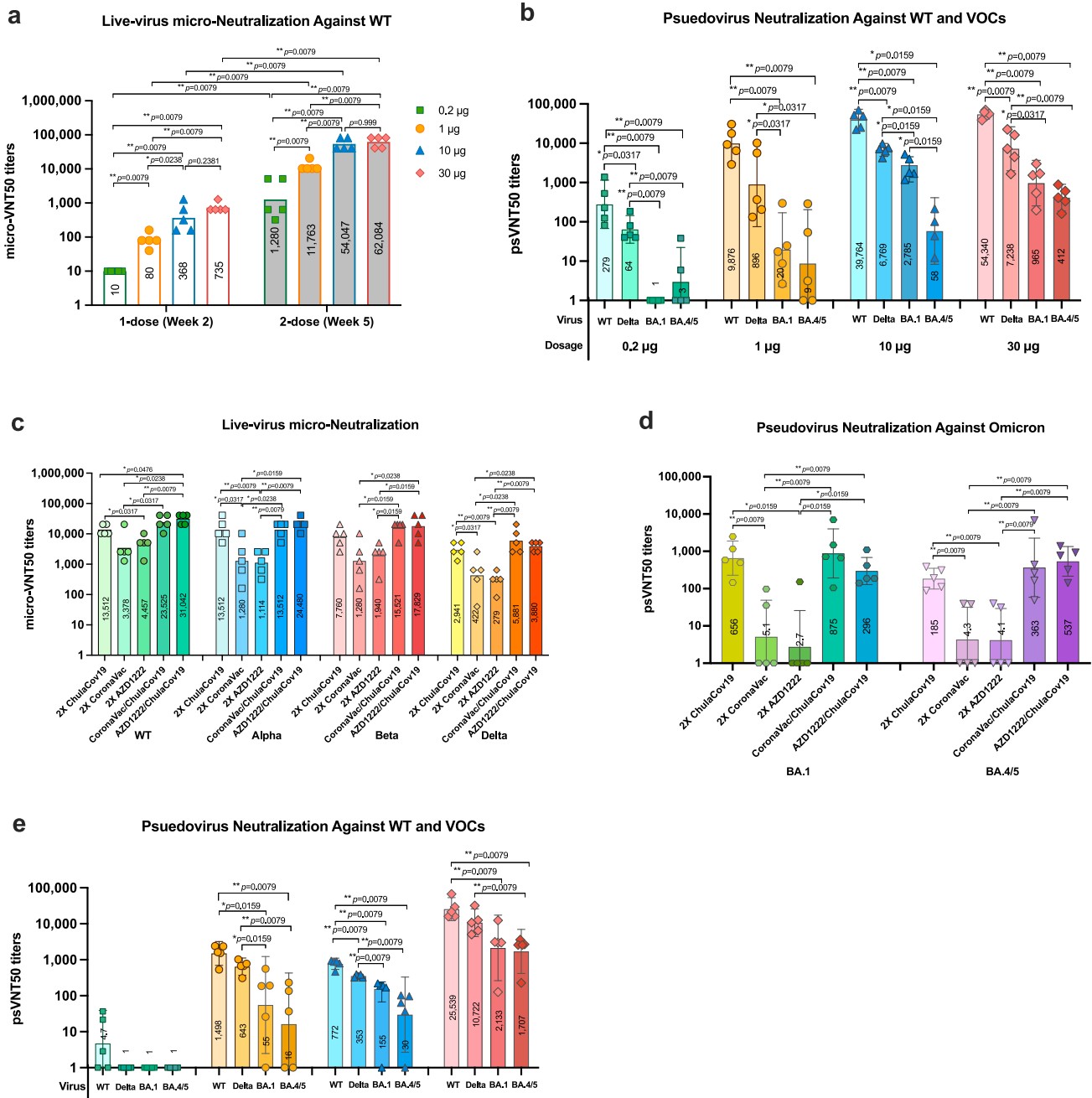

**Fig. 4 | NAb responses in immunized mice.** Experiment 1: (**a**) Live-virus micro-neutralization (micro-VNT50) titers against WT (Wuhan-Hu1) live-virus at two weeks after receiving each vaccine dose. **b** Pseudovirus neutralization test (psVNT50) titers at two weeks after the second dose against WT (Wuhan-Hu1), Delta (B.1.617.2), Omicron (BA.1, and BA.4/5) variants. Experiment 2: **c** micro-VNT50 titers against WT (Wuhan-Hu1), Alpha (B.1.1.7), Beta (B.1.351), and Delta (B.1.617.2) live-virus at two weeks after receiving various homologous or heterologous prime/boost regimens. **d** psVNT50 NAb titer results at two weeks after the second dose in various prime/boost regimens against Omicron BA.1 and BA.4/5 subvariants.

Experiment 3: **e** psVNT50 NAb against WT (Wuhan-Hu1), Delta (B.1.617.2), and Omicron (BA.1 and BA.4/5) variants for NAb durability and effect of 3rd dose of ChulaCov19 studies. The titers were determined in duplicate assays from 5 mice in each group. Note; 4 mice in 10 μg group were analyzed for psVNT50 against BA.4/5 due to the limited volume of serum samples. Each dot represents an individual animal. Bars represent the GMTs and 95% CI for each group. Statistical significance was determined by two-sided Mann–Whitney tests. Differences were considered significant at $p < 0.05$ with exact $p$-values shown. $p < 0.05$ and $p < 0.01$ are indicated by * and **, respectively. Source data are provided as a Source Data file.

pathology in adjuvanted recombinant RSV immunized mice[45]. Thus, in this study, vaccine-induced disease enhancement is less likely as demonstrated by the Th1-oriented response (Fig. 3b).

The induced NAb was highly specific to the original variant, however, cross-neutralization against the VOCs was also observed. As expected, Omicron subvariants, especially BA.4/5, showed the largest drop in micro-VNT50 titers (Fig. 4b). This was concordant with the

previous findings that Omicron subvariants could evade NAb induced by the first-generation or WT-virus-based vaccines[46]. The NT50 titer decrease found in our study was similar to those of other approved vaccines as the titers against BA.1 and BA.4/5 decreased by more than 8-10 folds when compared to the WT virus[46–48]. Therefore, during the surge of Omicron globally, there is a need of a boosting dose even with a first-generation vaccine or ideally with a second-generation vaccine

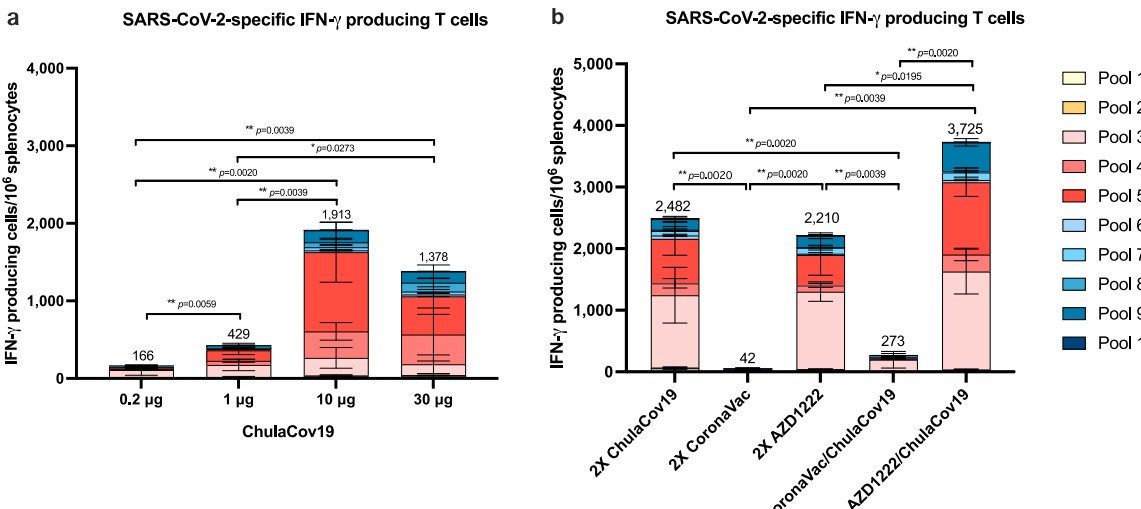

**Fig. 5 | Induction of S-specific IFN-γ positive T cells in BALB/c mice. a** mice were immunized with various doses of ChulaCov19 analyzed at 2 weeks after the second dose. **b** heterologous prime/boost study; mice were primed with CoronaVac or AZD1222 vaccine and boosted with ChulaCov19 (5 μg). Homologouse prime/boost results of each vaccine were included. S-specific IFN-γ positive T cells were determined in duplicate assays from 5 mice in each group. Bars represent the mean ± SD of S-specific IFN-γ positive T cells after stimulated with overlapping peptide pools spanning the SARS-CoV-2 S1 (pooled #1-5) and S2 (pooled #6-10). Statistical significance was determined by two-sided Mann–Whitney test. Differences were considered significant at $p < 0.05$ with exact $p$-values shown. $p < 0.05$ and $p < 0.01$ are indicated by * and **, respectively. SD; standard deviation. Source data are provided as a Source Data file.

such as a bivalent immunogen containing or encoding of Omicron's spike protein[49,50].

Two approved mRNA vaccines, Comirnaty™ by Pfizer/BioNTech and Spikevax™ by Moderna, comprise 2 proline substitutions at residues 986 and 987 of the S-protein (known as S-2P) to stabilize the prefusion conformational structure. However, it has not been shown that COVID-19 mRNA vaccine encoding non-stabilized spike protein is not immunogenic or is not protective against viral challenge. In these preclinical studies in mice, we have demonstrated that ChulaCov19, a secreted, prefusion non-stabilized ectodomain spike mRNA vaccine, elicited robust Spike-specific antibody and T-cell responses which has also translated into efficacy in protecting transgenic mice from SARS-CoV-2.

In many countries, immunization regimens have frequently employed mixtures of different vaccine platforms (also known as a heterologous prime-boost). This is especially true of the mRNA vaccines, and the approach has shown better results than homologous prime-boost with a non-mRNA-based vaccine[51]. In mice, ChulaCov19 was highly immunogenic as a booster in settings primed with either inactivated or viral vector vaccine. ChulaCov19 significantly enhanced the magnitude of both NAb and T cell responses compared to homologous 2-dose regimens of either CoronaVac or AZD1222. The NT50 titers against WT and Delta variants increased 7- to 14-fold when using the heterologous approach with ChulaCov19 as compared to the homologous immunizations with CoronaVac or AZD1222 (Fig. 4c). In the clinical setting, >8 weeks interval for AZD1222 was recommended to maximize the vaccine efficacy[52]. Hence, the low micro-VNT50 titer in the homologous AZD1222 group might increase if the interval between each dose is longer than 4 weeks as used in this study. In terms of spike-specific T-cell responses, our study found that AZD1222 prime/ChulaCov19 boost induced the highest magnitude of T cell response, superior to that of all tested regimens, including the homologous ChulaCov19 (Fig. 5b). This observation correlates with that of a recent clinical study report[53].

As the Omicron subvariant BA.4/5 is currently spreading worldwide, we have also assessed cross-neutralization and found that the NAb GMT measured by psVNT50 against BA.4/5 in homologous ChulaCov19 vaccination or heterologous boosted with ChulaCov19 groups were significantly better than either of the CoronaVac or AZD1222 homologous vaccination (Fig. 4d). This is consistent with a previous report[46].

The vaccine inequity issue is a huge challenge to healthcare in LMICs. In all past pandemics, as well as the ongoing one with COVID-19, access to effective vaccines in a timely manner and has been severely limited in these countries. The most effective COVID-19 vaccines are mRNA-based and were first approved in the United Kingdom, the United States, and Europe. They were widely available in these countries for approximately a year before being accessible on other continents. LMICs received these vaccines much later and in shorter supply, as evidenced by the most recent statistic (as of 31 August 2022) that in several African countries less than 30% of the population has received at least one vaccine dose[20]. Developing mRNA vaccine technology for distribution in these regions is therefore extremely important[21].

The ChulaCov19 vaccine development program has exactly this goal, striving to address the current and future pandemics in LMICs[54]. The program is funded by the Government of Thailand. The promising preclinical study results presented here demonstrate that ChulaCov19 is highly immunogenic with protective efficacy. This candidate vaccine has now completed non-clinical toxicity and biodistribution studies and has entered Phase 1 and 2 human trials. More importantly, in partnering with a domestic vaccine manufacture, BioNet Asia, ChulaCov19 can now be manufactured and formulated locally[54]. This initiative is ready to be part of the global effort to make mRNA vaccines more quickly and widely available when facing new variants or the next pandemic.

The limitation of this study includes the limited samples for tissue viremia after challenge. Moreover, the tissue slides were examined unblind. In-depth investigation of viral burden in different tissues as well as T cells quality induced by various vaccine dosages are still required. These factors might make it difficult to draw a strong conclusion on vaccine efficacy from the current of experiments.

In summary, this mRNA vaccine development is an effort to set up the technology platform in LMICS. Here we demonstrated that an LNP-encapsulated mRNA encoding a secreted form of prefusion non-stabilized ectodomain of SARS-CoV-2 spike protein "ChulaCov19" was

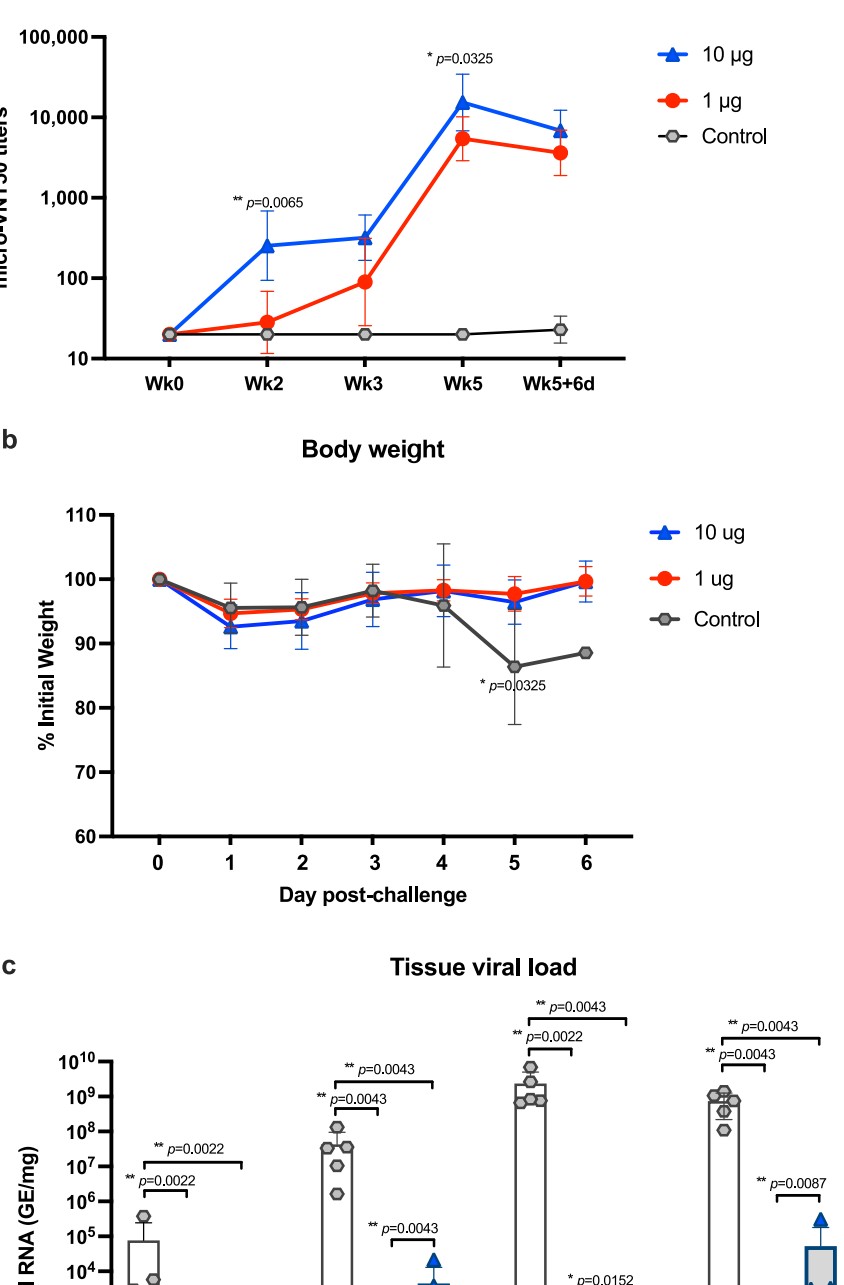

**Fig. 6 | Immune response and protective efficacy results in the challenge study.** **a** Kinetic response of micro-VNT50 titer after ChulaCov19 immunization and after challenge. The titers were determined in duplicate assays from control ($n = 5$) or vaccinated groups ($n = 6$), respectively. Data are presented as GMT of micro-VNT50 titer with 95% confident interval. Statistical analysis was performed to compare the GMT of micro-VNT50 between 1 and 10 µg dosed mice at each time point. **b** Body-weight values with SD are presented as a percentage of initial body weight before challenge (Day 0) through Day 6 post-challenge. In the control group, 3 out of 5 mice reached euthanasia criteria on Day 5 hence only 2 mice were analyzed for body weight on Day 6 after challenge. **c** SARS-CoV-2 viral RNA copies with SD detected by RT-qPCR in serum and homogenized tissues of challenged animals analyzed at euthanasia date (Day 6). Each dot represents an individual animal. Note: tissues from 3/5 animals in control group were collected at day 5. RNA copies were calculated as genomic equivalent/mg of tissue. Statistical significance was determined by two-sided Mann–Whitney test. $p < 0.05$ and $p < 0.01$ are indicated by * and **, respectively. SD; standard deviation. Source data are provided as a Source Data file.

**Table 1 | In-situ hybridization results of viral RNA in tissues**

| Vaccine Dose | Mice No. | Nasal Epithelium | Lung |
|---|---|---|---|
| 1 µg | 1 | 0 | 0 |
| | 2 | 0 | 0 |
| | 3 | 0 | 0 |
| | 4 | 1 | 0 |
| | 5 | 0 | 0 |
| | 6 | 0 | 0 |
| 10 µg | 1 | 0 | 0 |
| | 2 | 0 | 0 |
| | 3 | 0 | 0 |
| | 4 | 0 | 0 |
| | 5 | 0 | 0 |
| | 6 | 0 | 0 |
| PBS | 1 | 0 | 3 |
| | 2 | 1 | 4 |
| | 3 | 1 | 4 |
| | 4 | 1 | 3 |
| | 5 | 1 | 4 |

Note: Scores indicate subjective evaluation of the distribution of positive cells according to the following criteria: 0=negative; 1 = <10% positive; 2 = 10-25% positive; 3 = 25-50% positive; 4 = 50-75% positive; 5 = >75% positive.

able to elicit robust, specific antibody and T-cell responses. Chula-Cov19 vaccination could provide 100% protection from severe clinical signs and mortality in mice. It also markedly reduced viral RNA burden in serum and tissues. With such promising results from animal studies, the same formulation of ChulaCov19 vaccine that had been tested in animals is currently in phase 1-2 of clinical trials and can be manufactured locally for later clinical development. This program is a strong foundation for the fight against the next pandemic by increasing preparedness to make mRNA vaccine widely and timely accessible for LMICs, including Thailand.

## Methods

### Ethics statement
The investigators strictly adhered to the principles and guidelines of the Institute of Animals for Scientific Purposes Development, National Research Council of Thailand. All studies were conducted under protocols approved by the Committees on Care of Laboratory Animal Faculty of Medicine, Chulalongkorn University (IACUC approval no. 007/2563), and the Armed Forces Research Institute of Medical Sciences, AFRIMS (IACUC approval no. PN20-06).

### Cells and viruses
Vero E6, green monkey kidney epithelial cell line, was obtained from ATCC (Old Town Manassas, VA, USA). HEK293T-hACE-2 cells, prepared by transduction of HEK293T cell line with lentiviral habouring hACE-2 gene, used for hACE-2 binding assay was gratefully provided by Dr.Navapon Techakriengkrai[55]. Vero E6 and HEK293T-hACE-2 were grown in Eagle's minimum essential medium (EMEM) and Dulbecco's Modified Eagle's Medium (DMEM), respectively supplemented with 5-10% heat-inactivated fetal bovine serum (HIFBS), 1% L-glutamine, 1% Pen/Strep, 40 µg/ml gentamicin and 0.25 µg/ml fungizone (all were from Invitrogen, Carlsbad, CA, USA) at $35 \pm 2$ °C with 5% $CO_2$. One-day-old Vero E6 cells were used for measuring the level of neutralizing antibodies by live-virus micro-neutralization (micro-VNT50). The information of SARS-CoV-2 isolates including, wild-type (Wuhan-Hu1), Alpha (B.1.1.7), Beta (B.1.351) and Delta (B.1.617.2) variants for micro-VNT50 assay performed at the Department of Microbiology, Faculty of Sciences, Mahidol University was described previously[56,57]. For SARS-

CoV-2, Wuhan lineage (Hong Kong/VM20001061/2020, NR-52282) used for micro-VNT50 that performed at AFRIMS was obtained through BEI Resources (NIAID, USA). Viruses were propagated in Vero E6 cells to generate sufficient titers 100TCID_{50} for the micro-VNT50 assay. All isolates were quantitated by tissue culture infectious dose $TCID_{50}$ using the Reed-Muench method.

### Plasmid construction
Human codon-optimized sequences of the ectodomain of SARS-CoV-2 spike protein, amino acid position 1-1,210 (Wuhan Hu-1 complete genome, GenBank: MN908947.1, https://www.ncbi.nlm.nih.gov/nuccore/MN908947.1) was synthesized by GenScript, Piscataway, NJ, USA). Detailed amino sequence was shown in Supplementary File 1. It was subcloned into pUC-ccTEV-A101 using *Afe* I and Spe I restriction sites[58]. The plasmid was propagated in *E. coli* (Stbl3™, Invitrogen, Carlsbad, CA, USA) and extracted by EndoFree® Giga Kit (Qiagen, Hilden, Germany).

### In vitro transcription and mRNA encapsulation
Nucleoside-modified mRNA was produced by in vitro transcription (IVT) by substitution of uridine triphosphate (UTP) with N1-methylpseudouridine (m1Ψ) triphosphate (TriLink, Biotechnologies, San Diego, CA, USA), detailed elsewhere[58]. The reaction was carried out employing T7 RNA polymerase (MegaScript, ThermoFisher Scientific, MA, USA) on a linearized plasmid (*Not* I/*Afl* II double digestions). The mRNA was transcribed to contain 101 nucleotide of adenine (101-poly(A) tails). mRNA capping was performed by the trinucleotide cap1 analog, CleanCap (TriLink Biotechnologies, San Diego, CA, USA). The capped mRNA was purified by cellulose columns purification[59]. IVT mRNA was analyzed on agarose for determination of its integrity. Additional quality control to ensure the absence of double-stranded RNA (dsRNA) and endotoxin contamination prior to encapsulation into lipid nanoparticles (LNPs) were performed as described previously[60]. mRNA encapsulation was performed by Genevant Sciences Corporation (Vancouver, British Columbia, Canada). The proprietary lipid and LNP composition are described in patent application WO2020097540A1[61,62]. The LNP-encapsulated mRNA were characterized for their size, polydispersity using a Zetasizer (Zetasizer Nano DS, Malvern, UK), encapsulation efficiency, and shipped on dry ice and stored at −80 °C until use. The particles were re-characterized at 6- and 12-month after manufacture for stability assessment.

### mRNA transfection and in vitro protein expression analysis
At 24 h before transfection, $1 \times 10^5$ Vero E6 cells were seeded in a 24-well plate (Thermo Fisher Scientific, MA, USA). Cells with approximately 80−90% confluency were transfected with 1 µg of IVT ChulaCov19 using Lipofectamine™ MessengerMax™ (Invitrogen, Carlsbad, CA, USA) according to the manufacturer protocols. At 24 hr post-transfection, both intracellular and secreted S protein expressions were analyzed. For intracellular analysis, cells were fixed, permeabilized with ice-cold acetone and stained with 1:200 dilution of monoclonal-anti-RBD (R&D Systems, MN, USA), polyclonal-anti-S1, -anti-S2 antibodies (Sino Biological, Beijing, China), or 1:5,000 dilution of pooled convalescent serum (PCS) collected in 2020. Goat-anti-mouse IgG-FITC, donkey-anti-rabbit IgG-FITC (both were from BioLegend, CA, USA) or goat-anti-human AlexaFluor647 (Southern Biotech, AL, USA), at dilution of 1:5,000 was used as secondary antibodies following anti-RBD, -S1, -S2 or PCS staining. Secreted S protein was also subjected for analysis of its binding capability to hACE2. Supernatant collected from transfected cell was incubated with HEK293T-hACE-2 at 37 °C for 1 h then washed twice with PBS. Cells were then fixed with 4% paraformaldehyde for 30 min at RT. S protein on HEK293T-hACE-2 was stained with anti-RBD, -S1, -S2 or PCS and detected using the same procedure described above. Cell nuclei were counter stained with 4, 6-diamino-2-phenylindole

hydrochloride (DAPI) (Sigma-Aldrich, USA). Stained cells were visualized under confocal microscope (ZEISS LSM 800, Carl Zeiss, Germany). For western blot analysis, cell culture supernatant was analyzed by 12% polyacrylamide gel then transferred onto nitrocellulose membrane. Monoclonal anti-RBD (1:2,500), polyclonal-anti-S1 (1:5,000), -anti-S2 (1:5,000) or PSC (1:5,000) were used for detection of S protein in this step. Goat-anti-human IgG, goat-anti-mouse IgG, or goat-anti-rabbit IgG antibodies (all were diluted 1:10,000) conjugated with horseradish peroxidase (HRP) were used as secondary antibodies (all were from KPL, MD, USA) and detected by chemiluminescence substrate (Immobilon western, Millipore, CA, USA) then exposed to an X-ray film. Recombinant S protein with S1/S2 cleavage site abolished (ACROBioSystems, China) was used as positive control both in HEK293T-hACE-2 binding assay and western blot.

### Immunization in BALB/c mice

To address dose-response study of the ChulaCov19 and heterologous prime-boost responses with other approved COVID-19 vaccines, female BALB/c mice (*Mus musculus*), 4-6 weeks of age, (procured from Nomura Siam International, Bangkok, Thailand) were randomly divided into 5 mice/group for 3 sets of experiment. Experiment 1: dose-response of homologous ChulaCov19 prime/boost study, mice were immunized twice intramuscularly at 3 weeks interval of ChulaCov19 with dosage ranging from 0.2, 1, 10, to 30 µg. Experiment 2: a prime/boost regimen of 5 µg of ChulaCov19 and 1/10 of human dosage of approved vaccines available during the study period, including viral-vectored (ChAdOx1; AZD1222, Lot A10062, Nonthaburi, Thailand) and inactivated (CoronaVac, Lot C202105081, Beijing, China) vaccines. Five micrograms of ChulaCov19 was selected as we aimed to standardize the dosage to 1/10 of human dose for all vaccines (50 µg per dose of ChulaCov19 was used in phase II studies, Clinical Trial Identifiers: NCT05231369 and NCT05605470)[63,64]. In the homologous prime/boost of these 2 approved vaccines groups, each was given at four weeks interval. The 4-week gap was used according to the preclinical study protocol of ChAdOX-vectored vaccines[65,66]. The goal of experiment 2 was to assess the potential role of ChulaCov19 as a booster in a setting of heterologous primed with other COVID-19 vaccine platforms. Additional group (Experiment 3) immunized with 5 µg of ChulaCov19 was included for evaluation of NAb durability as measured at week 18 (15 weeks after received the 2nd dose) and the boosting effect of 3rd ChulaCov19 dose administered at week 20. Mice were bled at 2 weeks after each dose and antibody responses were measured by ELISA and/or neutralization assays. Splenocytes were collected at 2 weeks after the last dose (Experiment 1 & 2) for assessment of spike-specific IFN-γ T-cell using ELISpot assay (Fig. 1a). During the experiments, mice were maintained at 20–22 °C and a relative humidity of 45 ± 10% on a 12 h light/dark cycle.

### Challenge study in K18-hACE2 transgenic mice

Challenge study was conducted in ABSL-3 facility at AFRIMS, Bangkok, Thailand. Seventeen female K18-hACE2 mice (B6.Cg-Tg(K18-hACE2)2Prlmn/J), 7 weeks old (The Jackson Laboratory, Bar Harbor, ME, USA) were randomly divided into 3 groups. For group 1 and 2, there were 6 mice/group immunized intramuscularly via quadricep muscles with 2 doses, 3 weeks apart of ChulaCov19 at dose of 1 µg and 10 µg, respectively. In negative control (group 3), 5 mice were immunized with PBS instead of ChulaCov19 using the same schedule. At 2 weeks after the second immunization, mice were challenged intranasally with $2 \times 10^4$ pfu (in 50 µL) of SARS-CoV-2 (wild-type). Blood was collected at wk0, wk2, wk3, wk4 + 6 and wk5 + 6 days for antibody kinetic analysis (Fig. 1b). Six-day post challenge, wk5 + 6 days, mice were sacrificed to determine virus titers in different tissues (nasal turbinate, brain, lung, and kidney) and for histopathology. Virus titers were quantified by RT-qPCR and by determined the $\log_{10}TCID_{50}$ values. K18-hACE2 mice were

also housing at 20–22 °C and a relative humidity of 45 ± 10% on a 12 h light/dark cycle.

### Immunogenicity measurements

**Total IgG and IgA by ELISA.** S-specific IgG measurement was performed employing indirect ELISA as described previously[56,67]. In brief, 100 ng of recombinant S-trimer (ACROBioSystems, China) was coated to the 96-well plates. The 5-fold serially diluted mice sera were added in duplicate. After 1 h incubation at 37 °C, plates were washed vigorously with washing buffer (PBS + 0.5% Tween 20, PBST). Then, HRP-conjugated secondary antibodies, including rabbit anti-mouse IgG, dilution 1:10,000 (KPL, MD, USA), -IgG1 (dilution 1:5000), or -IgG2a dilution 1:5000 (both were from Southern Biotech, AL, USA) were added for an additional 1 h. After washing, the signals were detected by adding tetramethylbenzidine (TMB) substrate (BioLegend, San Diego, CA, USA). The reactions were then stopped with 50 µL of 0.16 N sulfuric acid. The absorbance was measured at a wavelength of 450 nm using a Varioskan microplate reader (ThermoFisher Scientific, Vantaa, Finland). Mid-point titers were calculated and expressed as the reciprocals of the dilution that showed an optical density (OD) at 50% of the maximum value substracted with the background (BSA plus secondary antibody).

Slightly different protocol in analyzing the presence of anti-SARS-CoV-2 IgG and IgA antibodies in sera mice from the challenge experiment were employed at AFRIMS. Briefly, 100 ng/well of RBD recombinant proteins (Abcam, UK) were coated overnight to the 96-well plates. Heat-inactivated mice sera that were diluted 1:100 was added in duplicates into RBD-coated wells and incubated at RT for 2 h. Then, either goat-antimouse IgG-HRP (1:40,000 dilution, KPL, USA) or goat-anti-mouse IgA-HRP (1:10,000 dilution, KPL, USA) was added to each well (100 µl/well) and incubated at RT for 1 h. The peroxidase reaction was visualized by adding Sureblue TMB solution (KPL, USA) and incubating in the dark at RT for 15 and 20 min for IgG and IgA ELISAs, respectively. The reaction was stopped by adding 50 µl/well of 0.5 M sulfuric acid. Absorbance at 450 nm was determined with a spectrophotometer. The $OD_{450}$ of blanks were subtracted from $OD_{450}$ of each sample before calculating antibody titer. The positive cut-off was the subtracted $OD_{450}$ + 3 SD.

**Neutralizing antibody.** In the immunogenicity dose-response and prime/boost studies (Experiment 1 and 2), NAb measurement was carried out as previously described[56,68] based on live-virus micro-VNT50 against WT (Wuhan-Hu1), Alpha (B.1.1.7), Beta (B.1.351), Delta (B1.617.2) variants in VERO E6 cells with positive cut-off of 1:20. In addition, the pseudovirus neutralization test (psVNT50) against lentiviral pseudovirus bearing a codon-optimized spike gene, described previously[69,70], was also used for determination of the neutralizing activity against WT, (Wuhan-Hu1), Alpha (B.1.1.7), Beta (B.1.351), Delta (B1.617.2), and Omicron (B1.1.529; BA.1 and BA.4/5 subvariants) variants. In the challenge study, NAb was also assessed by live-virus microneutralization test against strain hCoV-19/Hongkong/VM20001061/2020 with slightly different incubation period and detection technique. In the latter VNT protocol, serum-virus mixtures were incubated in VERO E6 cells for 5 days. In the detection step, staining of the living cells with 0.02% neutral red (Sigma Aldrich, USA) in 1X PBS (Invitrogen, Carlsbad, CA, USA) was used instead of viral protein staining employing anti-nucleocapsid (1:5,000) used in Experiment 1. Lysis solution was added for 1 h at RT before measuring OD at 540 nm. Percentage of virus infectivity in virus control (VC) and samples were calculated based on OD of cell control (CC), infectivity (%) = (OD of CC − OD of sample) x 100. The micro-VNT50 titers was calculated as the reciprocal serum dilution that neutralized 50% of virus observed in virus control wells using probit analysis, SPSS program[71].

**SARS-CoV-2-spike specific IFN-γ-producing T-cell measurement.** The procedure of mouse IFN-γ ELISPOT used in this study was described in our previous reports[56,72]. In brief, mouse splenocytes at $5 \times 10^5$ cells/well were cultured with SARS-CoV-2 spike peptide pools spanning the entire sequence of spike protein, 25 peptides/pool (Mimotopes, Mulgrave, Victoria, Australia) at a final concentration of $2 \mu g/mL$ at 37 °C, 5% $CO_2$ for 40 h. Pools 1–5 and 6–10 corresponded to S1 and S2 regions of spike protein, respectively. Secreted mouse IFN-γ was captured by anti-mouse IFN-γ (AN18) monoclonal antibody at dilution of 1:2,500 (Mabtech, Nacka Strand, Sweden) precoated on 96-well nitrocellulose membrane plates (Merk Millipore, Darmstadt, Germany). Results were expressed as spot-forming cells (SFCs)/$10^6$ splenocytes after subtraction of the spots from negative control wells.

**Real-time PCR for quantification of SARS-CoV-2 RNA.** SARS-CoV-2 RNA levels in serum and tissue samples were quantitated using quantitative RT-PCR. Viral RNA was extracted from 140 μl serum and tissue samples using the QIAamp viral RNA mini kit (QIAGEN, Hilden, Germany). For tissue samples, RNeasy Mini Kit (QIAGEN, Hilden, Germany) was used following manufacturer instructions. The total volume of 50 μl of viral RNA was obtained from each sample. Five microliters of each RNA sample was used in quantitative RT-PCR that was performed using CDC procedure[73] and AFRIMS SOPs in vitro SARS-CoV-2 RNA transcripts (IVTs). In each experiment, 3 internal controls (No Template Control (NTC), Negative Extraction Control (NEC) and Positive Extraction Control (PEC)) and 6 in vitro transcribed RNA standards were run along with test samples in each experiment. The number of copies of viral RNA per sample was derived from standard curves of serial dilutions of IVTs ($5, 50, 5 \times 10^2, 5 \times 10^3, 5 \times 10^4, 5 \times 10^5$ RNA copies number or genomic equivalent (GE)/reaction were included. The GE/ml of virus in a serum sample was calculated by multiplying the number of copies/reaction by [10,000 x the volume of a serum sample used (μl) for extraction]. The GE per gram of virus in a tissue sample was calculated by multiplying the number of copies/reaction by [10,000 x the weight of a tissue sample (mg) used for extraction].

**In situ hybridization and histology.** To detect SARS-CoV-2 RNA localization in mouse tissues samples, FFPE tissues of lung and nasal cavity were performed by using RNAscope In situ hybridization (ISH) assay. A SARS-CoV-2 probe (RNAscope® Probe, V-nCoV2019-S, Advanced Cell Diagnostics ACD, Newark, CA (ACD, 848561)) was used. The RNAscope® ISH assay was performed using an RNAscope 2.5 HD Red Detection Kit (ACD, 322372) as followed. The FFPE tissue slides were deparaffinized and treated with hydrogen peroxide (10 min at room temperature) followed by target retrieval in 1X target retrieval solution in a steamer of at least 99 °C for 15 min. Slides were then incubated with protease plus for 20 min at 40 °C in a HybEZTM oven (ACD) and subsequently incubated with the SARS-CoV-2 specific probe for 2 h at 40 °C in the HybEZTM oven. The signal was amplified using a specific set of amplifiers (AMP1-6) as recommended by the manufacturer and was detected using a Fast Red solution for 5 min at room temperature. Slides were counterstained with 50% Gill hematoxylin III (Sigma Aldrich, St Louis, MO, USA) for 2 min and extensively washed under tap water. The slides were dehydrated in 60 °C dry oven until completely dry and then dipped in Xylene before mounting with a mounting medium. SARS-CoV-2 RNA-positive cells were examined and counted unblind by certified personnel. The score (0-5) was assigned according to the percent distribution of fluorescent-positive cells.

**Statistical analysis**
Statistical analysis was performed using GraphPad Prism 9.0 software (San Diego, CA, USA). Comparisons of the data between groups were made using non-parametric tests (Mann–Whitney test). All $p$ values <0.05 were defined as statistically significant.

**Reporting summary**
Further information on research design is available in the Nature Portfolio Reporting Summary linked to this article.

## Data availability
The data supporting the findings of this work are available within the paper and in the Supplementary Information file. Source data are provided as a source data file. Source data are provided with this paper.

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

## Acknowledgements
The authors acknowledge all the members of the Chula VRC for their input and support. The authors would like to thanks Dr.Navapon Techakriengkrai, Faculty of Veterinary Science, Chulalongkorn University for providing HEK293T-hACE-2 cells. CK, EP and KR were funded by the National Vaccine Institute (NVI), grant No. 2563.1/8 and 2564.1/4, National Research Council of Thailand NRCT. CK was also funded by emerging Infectious Diseases and Vaccines Cluster, Ratchadapisek Sompoch Endowment Fund (2021), Chulalongkorn University (764002-HE04), and the Second Century Fund (C2F), Chulalongkorn University and Ratchadapiseksompotch Fund. EP was also supported by Faculty of Medicine, Chulalongkorn University, grant No. RA-MF-28/64.

## Author contributions
E.P., C.K., D.W., and K.R.: study conception and design, E.P., K.T., and C.K.: data collection, A.T., A.J., K.R., K.P., T.P., M.R., D.W., and K.R.: analysis and interpretation of results, M.G.A., K.T., P.K., N.Y., P.P., S.B., S.M., T.H., R.I.E., W.W., T.T., K.L., and J.H.: reagent preparation and analysis, E.P., C.K., and K.R.: draft manuscript preparation. E.P., C.K., and K.R.: grant funding acquisition. All authors reviewed the results and approved the final version of the manuscript.

## Competing interests
DW, and MGA are named on patents that describe lipid nanoparticles for delivery of nucleic acid therapeutics, including mRNA and the use of modified mRNA in lipid nanoparticles as a vaccine platform. KR, DW, MGA, CK, EP, and SB are co-inventors of the submitted ChulaCov19 mRNA vaccine's Patent. KL and JH are employees of Genevant Sciences Corporation and are named on patent describing lipid nanoparticles. WW is an employee of BioNet-Asia, Co. Ltd. We have disclosed those interests fully to their affiliations, and we have in place an approved plan for managing any potential conflicts arising from licensing of the patents. The remaining authors declare no competing interests.

## Additional information

**Eakachai Prompetchara**[1,2,3], **Chutitorn Ketloy** ®[1,2,3] ✉, **Mohamad-Gabriel Alameh** ®[4], **Kittipan Tharakhet**[1,2], **Papatsara Kaewpang**[1], **Nongnaphat Yostrerat**[1], **Patrawadee Pitakpolrat**[1,2], **Supranee Buranapraditkun**[1,5,6], **Suwimon Manopwisedjaroen** ®[7], **Arunee Thitithanyanont**[7], **Anan Jongkaewwattana**[8], **Taweewan Hunsawong**[9], **Rawiwan Im-Erbsin**[10], **Matthew Reed**[10], **Wassana Wijagkanalan**[11], **Kanitha Patarakul**[1,3,12], **Teerasit Techawiwattanaboon**[1,12], **Tanapat Palaga**[1,13], **Kieu Lam** ®[14], **James Heyes**[14], **Drew Weissman**[4,16] & **Kiat Ruxrungtham**[1,3,15,16]

[1]Center of Excellence in Vaccine Research and Development (Chula VRC), Faculty of Medicine, Chulalongkorn University, Bangkok 10330, Thailand. [2]Department of Laboratory Medicine, Faculty of Medicine, Chulalongkorn University, Bangkok 10330, Thailand. [3]Integrated Frontier Biotechnology for Emerging Disease, Chulalongkorn University, Bangkok 10330, Thailand. [4]Division of Infectious Diseases, University of Pennsylvania Perelman School of Medicine, Philadelphia, PA 19104, USA. [5]Department of Medicine, Faculty of Medicine, Chulalongkorn University, Bangkok 10330, Thailand. [6]Thai Pediatric Gastroenterology, Hepatology and Immunology (TPGHAI) Research Unit, Faculty of Medicine, Chulalongkorn University, Bangkok 10330, Thailand. [7]Department of Microbiology, Faculty of Science, Mahidol University, Bangkok 10400, Thailand. [8]Virology and Cell Technology Research Team, National Center for Genetic Engineering and Biotechnology (BIOTEC), National Science and Technology Development Agency (NSTDA), Pathumthani 12120, Thailand. [9]Department of Virology, Armed Forces Research Institute of Medical Sciences (AFRIMS), Bangkok 10400, Thailand. [10]Department of Veterinary Medicine, USAMD-AFRIMS, Bangkok 10400, Thailand. [11]BioNet-Asia, Co. Ltd, Bangkok 10260, Thailand. [12]Department of Microbiology, Faculty of Medicine, Chulalongkorn University, Bangkok 10330, Thailand. [13]Department of Microbiology, Faculty of Science, Chulalongkorn University, Bangkok 10330, Thailand. [14]Genevant Sciences Corporation, Vancouver, BC V5T 4T5, Canada. [15]Department of Medicine, and School of Global Health, Faculty of Medicine, Chulalongkorn University, Bangkok 10330, Thailand. [16]These authors jointly supervised this work: Drew Weissman, Kiat Ruxrungtham.
✉e-mail: chutitorn.k@chula.ac.th

