## [Peer Review File · Nature Communications]

REVIEWER COMMENTS

Reviewer #1 (Remarks to the Author):

This manuscript reported the preclinical studies on a SARS-CoV-2 mRNA vaccine encoding the prefusion-unstabilized ectodomain of the spike protein encapsulated in lipid nanoparticles (LNP) "ChulaCov19". Homologous prime-boost immunization with ChulaCov19 in BALB/c mice induced superior neutralizing antibody and T cell responses than that of inactivated (CoronaVac) or viral vector (AZD1222) vaccine. A clear dose-effect was found in immunogenicity with higher doses tended to induce broader neutralizing antibodies against Delta and Omicron variants. Heterologous prime-boost immunization with ChulaCov19 substantially augmented autologous neutralizing antibody to the wildtype virus as well as cross-neutralizing antibody to BA.1 and BA.4.5. ChulaCov19 immunization also demonstrated protection against challenge of authentic wildtype SARS-CoV-2 virus with reduced tissue viral load in nasal turbinate, lung and brain.

The manuscript appeared to be interesting. However, similar preclinical and clinical results have been widely published and the results presented in the manuscript does not constitute a significant enough conceptual advance.

The authors should provide sufficient information and quality control measures on ChulaCov19. For example, the exact sequence of the prefusion-unstabilized ectodomain of the spike, the particle size, and encapsulation efficiency. Apart from antibody binding to the fixed intracellular ectodomain of the spike, secreted ectodomain into the culture supernatant should also thoroughly analyzed such as their binding capacity to receptor ACE2 and to various NTD, RBD, and perhaps S2-specific antibodies to ensure the expressed ectodomain meets the initial design.

Similar to those published studies elsewhere in experimental animals and in humans in both homologous and heterologous prime-boost settings, ChulaCov19 induced reasonably good neutralizing antibody and T cell responses. However, the critical questions remain as how long these good responses could last and how broad they could protect against SARS-CoV-2 variants particularly to Omicron BA.1 and/or BA.4/5. Unfortunately, none of these questions were addressed in this manuscript. The results on the levels of immunity and protection against wildtype viruses are all within the realm of published results.

Some careless mistakes were also found here and there.

Lines 395-398, "In the heterologous vs homologous prime/boost experiment (Experiment 2),.....IFN-positive T cells responses which was 2,482 and 1,899 SFC/million splenocytes, respectively". However, 1,899 SFC/million could not be found for 2x AZD1222 in Figure 5. Should that be 2,210 SFC/million splenocytes?

Lines 398-401, "the heterologous AZD1222-prime/ChulaCov19-boost induced the.....spike-specific IFN-positive T cells of 3,725 SFC/million splenocytes, which significantly higher than all groups except homologous ChulaCov19. ... (Figure 5B)". Based on the number presented in Figure 5B, the IFN-positive T cells induced by homologous ChulaCov19 was 2,482 SFC/million which was NOT higher than 3,725 SFC/million induced by AZD1222/mRNA-S.

Th2 responses to a vaccine are associated with immune-enhanced disease (RSV). This is often associated with increased immune cells in the lung. It appears you did not find that in your animal model. It would be wise to discuss the RSV literature and point out that even though you find a Th2 response, this is not linked to immune-enhanced disease.

Reviewer #2 (Remarks to the Author):

The paper by Ketloy and colleagues is reasonably well written and clear. The results show what we think an mRNA vaccine can do, and it seems to confirm what other people have seen. I don't see this paper as really moving the science forward; its more of repeating what other people have seen paper. The authors state how important it is to advance novel technologies in low and middle income companies, and therefore showing mRNA technology can work in mice is a reasonable first step. The paper does seem to follow the correct methodologies expected in the field. There are definitely areas of improvement. Here's my suggestions.

Introduction:

Looks good.

Methods:

Mostly fine. I need more details in Table 1 and Supplemental data table on how this was done.

Results:

Generally clear. Some issues with using two different names in the figures for the mRNA vaccine (chulaCov19?). The challenge data is only ok. Data in 5B. I'm not sure if this is compelling or not. In hamsters, protected animals gain weight. These animals only reached the same weight as pre challenge. Maybe this can be addressed in the conclusions by comparison to other mouse ACE2 transgenic models. I'm confused why there's more virus was in the challenge from the 10 ug group. Figure 5C, viral RNA (GE/mg). Do you have an explanation?

Table 1. was not very compelling. Lots of ND. Were these not run because of a lack of sample or because they didn't have time? It's actually not good that the 10 ug group had the most of these, particularly given my comment on 5C. Is there an explanation of why 10 ug didn't do as well?

Supplementary Table. Should just be dropped. I really don't have any ideas what this means and how it really translates into protection. It's poorly explained.

Another minor concern, the doses for the mRNA vaccine immunogenicity seem high for a mouse. 30 ug. Everything is stated clearly, but only the 0.2 and 1 ug levels seem reasonable for a mouse study. The remaining dose level seem excessive for a mouse.

Discussion:

I would write the conclusion section much differently. Paragraph 1, focus on the most important results of the study. There seems to be a reasonable starting paragraph in paragraph 3, but there should be more about the successes in the challenge study. Paragraph 2, focus first on how these results compare to other vaccine investigations using the ACE2 transgenic mice. Are the results similar, improved, or same as described? This seems to be missing.

Conclusion:

I would tighten up the language a bit more. State that the mouse results were successful and served as the basis for a phase I trial using (similar or exactly) the same materials used in the mouse study as reported.

Minor:

Some misspelled works. e.g. United Kingdom.

Reviewer #3 (Remarks to the Author):

this paper details the pre-clinical development of an mRNA vaccine (ChulaCov19) against SARS-CoV2 in Thailand. Immunology following ChulaCov in different dosing schedules (and

as use as a booster) was investigated in mice. This was compared with an inactivated or virally vectored vaccine.

A humanised mouse model was then used to investigate efficacy of the novel vaccine in a SARS-CoV2 challenge.

ChulaCov showed good efficacy in the mouse model and comparable immunology to licensed vaccines.

This is a vaccine that has been developed and could be manufactured in Thailand (an LMIC), an important factor when thinking about vaccine independence for LMICs in the future.

Generally, the manuscript is well written but there are some instances where points are unclear due to the language.

Data analysis and methods look good however there are a number of conclusions that do not seem to be supported by the data which need to be rectified.

Please see attachment for detailed notes

Response to reviewers: NCOMMS-22-36983

Reviewer 1:

1. The authors should provide sufficient information and quality control measures on ChulaCov19. For example, the exact sequence of the prefusion-unstabilized ectodomain of the spike, the particle size, and encapsulation efficiency.

Response:

- Amino acid sequence detail was mentioned in the revised manuscript (Line 156-157, 158-159) and in Supplementary file 1.
- Encapsulated-mRNA characterization and stability results were included in the method (Line 179-180), results (Line 362-366) and in Supplementary Table 1, respectively.

2. Apart from antibody binding to the fixed intracellular ectodomain of the spike, secreted ectodomain into the culture supernatant should also thoroughly analyzed such as their binding capacity to receptor ACE2 and to various NTD, RBD, and perhaps S2-specific antibodies to ensure the expressed ectodomain meets the initial design.

Response:

- Analysis of secreted ectodomain in supernatant by western blot employing anti-S1, anti-RBD and anti S2 was added in method (Line 201-208), result (Line 352-355), revised Figure 2C)
- Binding capability of secreted ectodomain S protein with hACE-2 was included in method (Line 140-145, 192-197), results (Line 355-361) and Figure 2B).

3. Similar to those published studies elsewhere in experimental animals and in humans in both homologous and heterologous prime-boost settings, ChulaCov19 induced reasonably good neutralizing antibody and T cell responses. However, the critical questions remain as how long these good responses could last and how broad they could protect against SARS-CoV-2 variants particularly to Omicron BA.1 and/or BA.4/5. Unfortunately, none of these questions were addressed in this manuscript.

The results on the levels of immunity and protection against wildtype viruses are all within the realm of published results.

Response:

- Thank you very much for the concern, we've included the results of additional experiment (Experiment 3) which monitored the NAb against WT and VOCs until Week 18 (15 weeks apart from 2nd dose). This was for NAb durability assessment. The Experiment 3 was also examined the boosting effect of 3rd dose in mice previously primed at 17 weeks ago (Figure 1A, Line 227-229 of methods, and Line 420-430 of results).

4. Lines 395-398, "In the heterologous vs homologous prime/boost experiment (Experiment 2),.....IFN-positive T cells responses which was 2,482 and 1,899 SFC/million splenocytes, respectively". However, 1,899 SFC/million could not be found for 2x AZD1222 in Figure 5. Should that be 2,210 SFC/million splenocytes?

Response:

- We apologize for this mistake. It was corrected to 2,210 SFC/million splenocytes (Line 447)

5. Lines 398-401, "the heterologous AZD1222-prime/ChulaCov19-boost induced the.....spike- specific IFN-positive T cells of 3,725 SFC/million splenocytes, which significantly higher than all groups except homologous ChulaCov19. ... (Figure 5B)". Based on the number presented in Figure 5B, the IFN-positive T cells induced by homologous ChulaCov19 was 2,482 SFC/million which was NOT higher than 3,725 SFC/million induced by AZD1222/mRNA-S.

Response:

- This statement was rewrite and corrected both in the result and in the abstract (Line 62-63, 449-451).

6. Th2 responses to a vaccine are associated with immune-enhanced disease (RSV). This is often associated with increased immune cells in the lung. It appears you did not find that in your animal model. It would be wise to discuss the RSV literature and point out that even though you find a Th2 response, this is not linked to immune-enhanced disease.

Response:

- Thank you very much for your constructive suggestion, we've included it in discussion with references (Line 580-586).

Response to reviewers: NCOMMS-22-36983

Reviewer 2

Methods:

1. Mostly fine. I need more details in Table 1 and Supplemental data table on how this was done.

Response:

- Table 1 is the result of *in situ* hybridization (ISH) for detection of viral RNA in tissues. The detail of ISH procedure was included in method section (Line 321-338) and Table 1 description.

Results:

2. Generally clear. Some issues with using two different names in the figures for the mRNA vaccine (chulaCov19?).

Response:

- Thank you very much for your suggestion, the vaccine name was changed to ChulaCov19 throughout the manuscript.

3. The challenge data is only ok. Data in 5B. I'm not sure if this is compelling or not.

Response:

- Not sure whether the reviewer would like to mention Figure 6B (body weight) instead of 5B or not. For clearer demonstration of the weight change, we've revised Figure 6B to plot the percent change from initial weight instead of weight (in gram).

4. In hamsters, protected animals gain weight. These animals only reached the same weight as pre challenge. Maybe this can be addressed in the conclusions by comparison to other mouse ACE2 transgenic models.

Response:

- Thank you very much for the constructive comment. The body weight of vaccinated mice was slightly decreased by days 1 and 2 after challenge and was gradually increased to the same levels as pre-challenge by day 6. This was similar to the previous studies that used either HFH4-hACE2 or K18-hACE2 transgenic mice as challenge models. This statement was addressed in discussion with references (Line 524-531).

5. I'm confused why there's more virus was in the challenge from the 10 ug group. Figure 5C, viral RNA (GE/mg). Do you have an explanation?

Response:

- We have included this issue and proposed the possible causes of this findings in the discussion (Line 539-555).

6. Table 1. Was not very compelling. Lots of ND. Were these not run because of a lack of sample or because they didn't have time? It's actually not good that the 10 ug group had the most of these, particularly given my comment on 5C. Is there an explanation of why 10 ug didn't do as well?

Response:

- We strongly agree with your concern. The ISH data marked ND because the olfactory bulb and/or retina were not present in those examined sections tissues. We finally decide to remove the results of olfactory bulb and/or retina as they contain ND (revised Table 1). This is to avoid the misinterpretation of these results.

7. Supplementary Table. Should just be dropped. I really don't have any ideas what this means and how it really translates into protection. It's poorly explained.

Response:

- Table S1 was removed.

8. Another minor concern, the doses for the mRNA vaccine immunogenicity seem high for a mouse. 30 ug. Everything is stated clearly, but only the 0.2 and 1 ug levels seem reasonable for a mouse study. The remaining dose level seem excessive for a mouse.

Response:

- As the Experiment 1 was the first *in vivo* study of ChulaCov19, so, we would like to screen whether the vaccine was immunogenic. That's why a quite high dose (30 ug) was used in the initial step. In

the later experiments (Exp2 & 3), the dosage was reduced to only 5 ug (1/10 human dosage) which is 1/10 of human dosage used in clinical trials. Dosage of 30 ug is no longer use in mouse study.

Discussion:

9. I would write the conclusion section much differently. Paragraph 1, focus on the most important results of the study. There seems to be a reasonable starting paragraph in paragraph 3, but there should be more about the successes in the challenge study. Paragraph 2, focus first on how these results compare to other vaccine investigations using the ACE2 transgenic mice. Are the results similar, improved, or same as described? This seems to be missing.

Response:

- Thank you very much for the constructive suggestion, we've reordered the discussion starting from the successes in challenge study. Comparison with other vaccines using ACE2 transgenic mice was also included (Line 516-523).

Conclusion:

I would tighten up the language a bit more. State that the mouse results were successful and served as the basis for a phase I trial using (similar or exactly) the same materials used in the mouse study as reported.

Response:

- The statement for translation of ChulaCov19 preclinical studies to phase I/II clinical trial using the same formulation was included (Line 654-657).

Minor:

Some misspelled works. e.g. United Kingdom.

Response:

- We apologize for the mistake; typo error was corrected (line 629).

Response to reviewers: NCOMMS-22-36983

Reviewer 3

1. Please be consistent with naming of vaccine – use of mRNA and “ChulaCov” are confusing. This is found in the text and in figures (4C – alpha, 5A and B).

Response:

- Vaccine name was revised to ChulaCov19 throughout the manuscript.

2. Line 53 – micro virus neutralisation is abbreviated to (micro-VNT) however, later the term MN50 is used in the text. Please be consistent.

Response:

- Thank you very much for the suggestion, micro virus neutralisation was abbreviated to (micro-VNT) throughout the manuscript.

3. The used of 5ug of ChulaCov in experiment 2 is not justified. The dose finding (experiment 1) uses 0.2, 1, 10 and 30ug and the challenge (experiment 3) used 1 and 10ug dose. Please explain why 5 ug was used.

Response:

- As the Experiment 1 was the first *in vivo* study, so, we used various doses ranges (0.2, 1, 10 and 30 ug) for initial evaluation of Chulacov19 immunogenicity.

- Dosage of 5 ug was used in the later experiments because we would like to standardize all vaccines (in Experiment 2 and 3) in mice to 1/10 of human dosage (50 ug was used in ChulaCov19 Phase 2 trial, ClinicalTrials.gov Identifier: NCT05231369, NCT05605470). This was also stated in method (Line 220-222).

4. Please address why experiments 1 and 3 used a 3 weeks interval whereas experiment 2 used a 4 week interval between doses of vaccine.

Response:

- As we included licensed vaccine CoronaVac and AZD1222 in Experiment 2, for standardization of immunization schedule, we adopted the longest gap (from AZD1222), that used 4-week interval between each dose. Additional explanation regarding the different time gap and references were included in the method (Line 224-225).

5. Please state if the people reading the histological slides were blinded to group. If not, please add this to the study limitations.

Response:

- The histological slides were examined unblind by trained personnel. This issue was stated as a limitation in method and discussion (Line 336-338, 645-646).

6. Line 328 should read “untransfected” instead of “transfected” when saying that cells were negative for fluorescence I think.

Response:

- We apologize for the mistake, it was changed to “untransfected” (Line 350)

7. Figure 2 – please provide details of S0 in the legend.

Response:

- Details of S0 is provide in result (Line 352), Figure 2 legend (Line 1033-1034).

8. Please address the mismatch between Th1/Th2 with IgG2a and IgG1 in lines 340-341.

Response:

- The mismatch was corrected (Line 375).

9. Figure 3A – statistics should be added (and reported in the text) for 2 weeks vs 5 weeks in order to state that “The second dose of ChulaCov strongly augmented the IgG antibody levels”.

Response:

- Statistical analysis comparing between weeks 2 vs 5 IgG titer was included in the text (Line 374-375) and in the revised figure 3A.

10. Figure 4A – again, statistics should be added (and reported in the text) for 2 weeks vs 5 weeks to say that “The Nab titres were drastically enhanced after the second dose”.
- Response:**
- Statistical analysis comparing between weeks 2 vs 5 NAb titer was included in the text (Line 384) and in the revised figure 4A.
11. Line 356 – the sentence is not corroborated by the data – 0.2ug dose showed that BA.4/5 is higher than BA.1.
- Response:**
- The sentence was revised (Line 392).
12. Lines 366-370 – The data do not support this statement for Omicron. Figure 4D does not show statistical significance for 2xcoronavac vs coronavac/chulacov for either BA.1 or BA.4/5. Please amend based on the data.
- Response:**
- Statistics were re-analyzed and showed that coronavac/chulacov-19 induced psVNT50 titer significantly higher than 2xcoronavac for both BA.1 and BA.4/5 (Line 410-416), figure 4D was revised accordingly.
13. Line 376-377. I don't think that comparing NT to WT to those to Omicron is valid. The assays were different, and the methods section states those differences. Also, authors do not present statistics to back this up.
- Response:**
- Thank you for the suggestion, the results were revised to not compare Omicron with WT as the method is different (Line 410-416).
 - Statistics were analysed and shown in figure 4D to indicate the difference psVNT50 titers against BA.1 and BA.4/5 between groups.
14. Lines 378 – 380 – the data do not support the statement that “the heterologous prime/boost regimen was more efficient (84-172 folds increase) in inducing cross-NAb against VOCs including BA.1 and BA.4/5 subvariants than homologous CoronaVac or AZD1222 immunization” - Figure 4D does not show statistical significance for 2xcoronavac vs coronavac/chulacov for either BA.1 or BA.4/5.
- Response:**
- Statistics were analysed and shown in figure 4D to indicate the difference psVNT50 titers against BA.1 and BA.4/5 between groups.
15. Lines 394 – It would aid understanding if you included the percentages responding to the pools here.
- Response:**
- The percentage of spots following pool#3-5 and pool#9 stimulation were included in the text (Line 442-444).
16. Line 402 – Data in Fig 5B do not support the statement that “Boosting with ChulaCov19 significantly enhanced the magnitude of T cells response in CoronaVac-primed mice”.
- Response:**
- The statement is revised (Line 453-454).
17. Line 415 – Does the p value relate to the difference between the 10ug and 1ug dose? Please clarify.
- Response:**
- The statement is revised to “At this time-point, 10 µg dosed mice induced significantly higher in GMTs of micro-VNT50 titers than 1 µg dosed mice ($p = 0.0065$)”. (Line 466-467).
18. Line 419 – does the p value relate to the difference between the doses at week 5? This is not clear.
- Response:**
- The statement is revised to “At week 5 (2 weeks after the second dose), all mice in both vaccinated groups showed an increase of NAb levels. The GMT of micro-VNT50 titers at week 5 were 15,343 and 4,424 in the 10 µg and 1 µg group, respectively, $p = 0.0325$ (Line 469-471).

19. Line 420 – was this decline significant?
Response:
- The decline of NAb was non statistically significant, included in text. “there was a slight decline of NAb titers in both groups but non statistically significant when compared to week 5, $p = 0.1126$ and $p = 0.4437$ for 10 μg and 1 μg groups, respectively.” (Line 472-473)
20. Line 423 – why are data not shown? Could they be included?
Response:
- The results were included as supplementary Figure S1A and S1B. Method for IgG and IgA in challenge study was also included (Line 262-273).
21. Line 431 – please clarify that the 5 mice were controls and received PBS.
Response:
- Yes, 5 control mice received PBS, mentioned in the text (Line 483 and figure 1B legend)
22. Line 436 onward – This section is confusing. Please amend Figure 6C to make serum the first plot on that figure. Please also include the actual GE/ml in serum for both vaccine doses since the levels are so low. Please remove the statement “indicating that vaccination leads to protection against SARS-CoV-2 infection” as there is clearly virus in nasal turbinates, lung and brain in Fig 6C.
Response:
- Figure 6C was revised and actual GE/ml in serum for both vaccine doses were included in the graph
- The the statement “indicating that vaccination leads to protection against SARS-CoV-2 infection” was removed.
23. Figure 6 legend. please clarify the p values (* and **) Figure 6B – Please spell out “standard deviation” and “body weight” and include what the statistics are calculated on. Figure 6C – please clarify the timepoint studied.
Response:
- Figure 6 legend was revised accordingly.
24. Line 466 – I think this should reference Figure 2A, not 3A.
Line 467 – Again, reference Figure 2B, not 3B.
Response:
- Figures were revised to Figure 2A, and 2C.
25. Line 468 – Suggest removing the 2 x references to “in nature”. This is in a cell line, not in nature.
Response:
“in nature” was removed
26. Line 475 – 477 – Please amend to reference immune responses elicited by AZD1222 which utilises the native-like structure.
Response:
- Ref #55 was added (Line 571-574).
27. Line 484 – please amend “B cell responses” to “antibody responses”. B cells were not studied here.
Response:
- B cell was changed to antibody response (Line 603).
28. Line 507 – Please amend “MN50” to micro-VNT, as already mentioned above. Additionally, there are no data here to support the statement “The low MN50 titer in the homologous AZD1222 group may have been influenced by anti-vector antibodies as the interval between doses was short (4 weeks).” Please amend.
Response:
- The statement was removed and revised to “Hence, the low micro-VNT50 titer in the homologous AZD1222 group might increase if the interval between each dose is longer than 4 weeks used in this study” (Line 615-616).

29. Line 510-511 – There are no data to here to support the statement “However, boosting with ChulaCov19 would avoid these concerns entirely, as well as shortening the vaccination gap and increase the speed of vaccine coverage”. Suggest removing.

Response:

- The statement was removed.

30. Lines 516-520 – again, data do not support this. There is no significance between 2 x coronavac vs coronavac and chulacov boost in Fig4D. Please amend.

Response:

- Additional statistical analysis in figure 4D indicated that CoronaVac-prime/ChulaCov19-boost induced significantly higher psVNT50 titers against BA.1 and BA.4/5 and figure 4D was revised.

31. Lines 526-529 – This statement “Of note, no SARS-CoV- protein was detected in organ tissues in mice vaccinated with ChulaCov19 at either the 1 or 10 µg dose” does not tally with the data. The data show evidence of virus in nasal turbinate, lung and brain (Fig 6A).

Response:

- Thank you for the suggestion, we revised to “Notably, SARS-CoV-2 RNA measured by ISH was undetected in lung tissue in mice vaccinated with ChulaCov19 at either the 1 or 10 µg dose. When RT-qPCR was used, although viral RNA was still detected in some tissues, both dosages demonstrated a 99-100% reduction of the viral RNA in tested tissues when compared to the control group. (Line 534-537)

32. Line 542 – suggest adding the date of the “recent” statistic

Response:

- Date added (Line 632).

33. I recommend discussing the VNT50 levels that are associated with protection in mouse models for other vaccines to put your findings in context (especially since this study did not include a licensed vaccine as a control in the challenge study).

Response:

- The challenge study of RNA1273 in K18-hACE-2 was compared and include in discussion (Line 558-561).

REVIEWERS' COMMENTS

Reviewer #1 (Remarks to the Author):

The authors have addressed all my concerns.

Reviewer #2 (Remarks to the Author):

The authors have done a good job addressing all my concerns.

Reviewer #3 (Remarks to the Author):

Thanks to the authors for the clear and thorough rebuttals and associated edits.
The inclusion of p values enables readers to more clearly understand the statistical significance of the findings.